# Effectiveness of suvorexant versus benzodiazepine receptor agonist sleep drugs in reducing the risk of hip fracture: Findings from a regional population-based cohort study

**Ryozo Yoshioka[1,2], Seiichiro Yamamoto[1], Eiji Nakatani[1]** *

**1** Graduate School of Public Health, Shizuoka Graduate University of Public Health, Aoi-ku, Shizuoka, Japan,
**2** Department of Emergency Medicine, Shizuoka General Hospital, Aoi-ku, Shizuoka, Japan

* nakatani.eiji.int@gmail.com

**Data Availability Statement:** According to Shizuoka Prefecture's data use agreement with local insurers, readers cannot access the analyzed

## Abstract

Sleep drugs are often necessary to treat insomnia in older patients. Benzodiazepine receptor agonists (BZRAs) are primarily used for insomnia in these patients, but there are concerns regarding their association with delirium and bone fractures. Among sleep drugs, orexin receptor antagonists such as suvorexant have a lower risk of delirium than BZRAs, but their effectiveness in preventing hip fractures is unknown. Hip fracture is a life-threatening trauma in advanced-age patients and a social problem. Therefore, we investigated the relationship between suvorexant and hip fracture. The Shizuoka Kokuho Database was used to compare the time to hip fracture in patients who had been newly taking suvorexant and other sleep drugs such as benzodiazepines since November 2014. A proportional hazards model for hip fracture as an outcome was used to estimate the hazard ratio. Propensity scores were estimated using a logistic regression model, and the confounding factors were age, sex, several comorbidities, and each oral medication. The suvorexant group comprised 6860 patients (110 with hip fracture), and the BZRA group (benzodiazepines and Z-drugs) comprised 50,203 patients (1487 with hip fracture). In the matched cohort (6855:6855 patients), 259 and 249 patients in the suvorexant and BZRA group developed hip fractures during the observational period, respectively. The hazard ratio of the suvorexant group compared with the BZRA group was 1.48 (95% confidence interval, 1.20–1.82). In the subgroup analysis, patients in the suvorexant group had a higher risk of hip fracture if they were aged >75 years, had no diabetes, had no neurological disease, had no renal failure, had liver disease, had hypertension, were not taking alpha 1 blockers, and were not taking oral steroids. Among people in the Japanese regional population who use sleep drugs, patients taking suvorexant can be at higher risk of hip fracture than patients taking BZRAs.

data. Researchers interested in accessing this data set may submit an application to Shizuoka Prefecture to request access. Please contact the staff of Shizuoka Graduate University of Public Health (e-mail: info@s-sph.ac.jp).

**Funding:** The Shizuoka Graduate University of Public Health conducts contract research projects for public health in Shizuoka Prefecture, including the current study, and funding for this work was provided by Shizuoka Prefecture. The funders had no role in study design, data collection and analysis, decision to publish, or preparation of the manuscript.

**Competing interests:** The authors have declared that no competing interests exist.

**Abbreviations:** BZRA, benzodiazepine receptor agonist; SKDB, Shizuoka Kokuho Database; NHI, National Health Insurance; LSMCSE, Late-Stage Medical Care System for the Elderly; ICD-10, International Classification of Diseases; 10th Revision, HR: hazard ratio; CI, confidence interval.

## Introduction

Insomnia is often a problem for people of advanced age because of its high incidence. In Japan, about 10% to 15% of the general population experiences insomnia [1], and prior studies have shown that the advanced-age population has a higher incidence of insomnia [2]. For this reason, sleep drugs are often prescribed to older patients. Benzodiazepine receptor agonists (BZRAs), such as benzodiazepine and Z-drugs (zolpidem, zopiclone, and eszopiclone), are widely used as sleep agents. However, BZRAs have side effects such as delirium and hip fractures. Hip fractures have attracted attention as a factor that increases the risk of death in today's aging society [3, 4]. In addition, once a hip fracture occurs, the patient is often bedridden and in need of nursing care [5], which is a significant problem from the perspective of extending healthy life expectancy. Previous studies have investigated risk factors for hip fractures caused by drugs [6–15], and the risk associated with BZRAs is particularly notable [11–15].

BZRAs (benzodiazepine and Z-drugs) bind to the ω1 benzodiazepine receptor and produce sleep and sedative effects. However, benzodiazepine receptors also include ω2 receptors, which create muscle relaxant and anxiolytic effects, and binding to these receptors is believed to cause lightheadedness and falls. Z-drugs have a weaker impact on ω2 receptors than does benzodiazepine, but side effects such as falls and lightheadedness remain. In contrast, suvorexant, launched in 2014 in Japan, promotes sleep onset by antagonizing orexin receptors (orexin receptors 1 and 2) and suppressing arousal without involving benzodiazepine receptors [16, 17]. This is because the drug inhibits the binding of orexin-to-orexin receptors, inhibiting over-acting arousal and thus allowing the brain to move into a sleep state.

The pharmacological effects described above suggest that unlike BZRA sleep drugs, orexin receptor antagonists do not have muscle relaxant effects and have a low risk of delirium because of their action of inhibiting wakefulness [18–20]. Recent studies have also indicated a preventive effect of suvorexant against delirium [19, 20]. Prior studies have shown an increased risk of fracture with delirium [21], and we hypothesize that these findings suggest that suvorexant is associated with a higher risk of fracture compared with BZRAs. However, because they are novel drugs, their clinical significance remains to be evaluated, particularly concerning falls and hip fractures. Because suvorexant is not a muscle relaxant and does not induce delirium, we investigated whether the orexin receptor antagonist suvorexant, a novel sleep drug, reduces the risk of hip fractures compared with BZRA sleep drugs.

## Materials and methods

### Database and Japanese medical insurance

This study was performed using the Shizuoka Kokuho Database (SKDB) [22], which is derived from a database that provides linked data such as demographic and registration data and medical claims for members of the Federation of National Health Insurance Association in Shizuoka Prefecture. With a population of approximately 3.6 million, Shizuoka Prefecture is a representative region in Japan in terms of climatic conditions and population distribution. The database mainly includes National Health Insurance (NHI) members (<75 years of age, approximately 25% of all prefectural residents) and beneficiaries of late-stage elderly healthcare (all prefectural residents ≥75 years of age).

The Japanese healthcare system is based on a comprehensive insurance system. There are two types of health insurance for people aged <75 years: Employee Health Insurance for the employees of government organizations and large companies and NHI for small business owners and their employees. Health insurance for people aged ≥75 years is provided by the Late-Stage Medical Care System for the Elderly (LSMCSE). The SKDB maintains data on the name

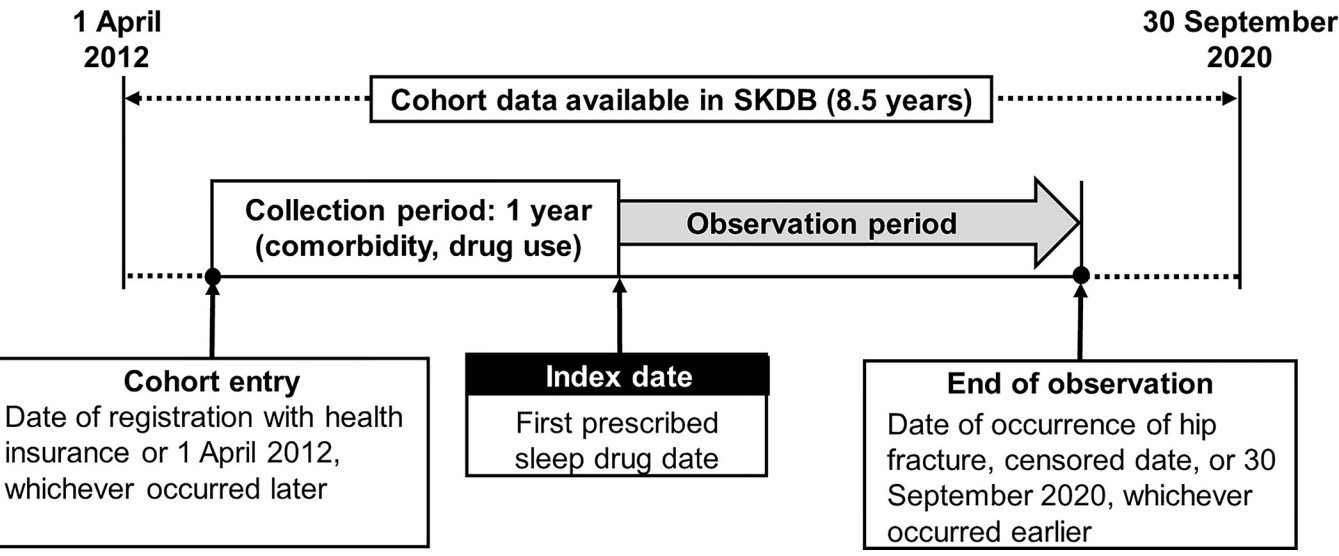

**Fig 1. Study schema.** SKDB: Shizuoka Kokuho Database.

of the prescribed drug, prescription date, prescription drug dosage, and number of prescription days.

## Study design and analyzed cohort

The study schema is shown in Fig 1. This population-based cohort study was conducted using the SKDB. The dataset comprised 8.5 years of longitudinal data from April 2012 to September 2020. All enrollees were investigated using the individually linked data in the databases for their insurance claims. Each enrollee's data availability period was defined as the time from the date of insurance registration or April 2012, whichever was later, to the date of insurance withdrawal or September 2020, whichever was earlier.

The study period extended from 1 November 2014, when suvorexant was launched, to 30 September 2020. The entry date of analyzed cohort participation was defined as the date of enrollment in corresponding health insurance organizations or 1 November 2014, whichever was later. The follow-up end date was determined as the date of the first diagnosis of hip fracture, the study end date (30 September 2020), or withdrawal from the NHI or LSMCSE, whichever was earlier.

The inclusion criterion was new prescription of sleep drugs during the above-described study period. The exclusion criteria were new prescription of sleep drugs before 2014, fracture outcome and sleep drug prescription occurring in the same month, and age of <65 years. We conducted this study with an intent-to-treat analysis.

## Sleep drugs

Patients were assigned to the BZRA group if they had received at least one BZRA sleep drug prescription (S1 Table) during the follow-up period and to the suvorexant group if they had received a suvorexant prescription during the follow-up period. For both groups, only the first drug administration was included in the analysis.

## Outcome and potential confounders

The outcome of this study was the incidence of hip fracture, ascertained from claims data using International Classification of Diseases, 10th Revision (ICD-10) codes (S720–722).

Age, sex, the Elixhauser comorbidity index, and covariates listed in previous reports were used as candidate confounders [9, 10, 23–27]. The Elixhauser comorbidity variable was determined to be present if the disease was identified using ICD-10 in the claims data [28], with a search period of 1 year before the date of the visit. Other comorbidities and medications that might be associated with the development of femoral fractures or drug selection were also defined (S2 Table). Their presence or absence was determined during the 1-year search period.

## Statistical analysis

Frequencies and percentages were calculated for categorical variables, and mean and standard deviation were calculated for continuous variables. Comparisons of continuous and categorical variables between the two groups were performed using the t-test and chi-squared test, respectively.

To predict the probability of patient allocation to the suvorexant group, a logistic regression model was used to estimate propensity scores using all prespecified covariates, and 1:1 matching was performed using the nearest-neighbor propensity score-matching method. After propensity score matching, the hazard ratio (HR) between the BZRA group and the suvorexant group was estimated using the Cox proportional hazards model. The model calculated the HR, the corresponding 95% confidence interval (CI), and univariables based on the Wald test. Before estimation of the HR, the proportional hazard assumption was checked. As an additional analysis, we estimated smooth-in-time parametric hazard functions to show the time-dependent risk after initiation of oral medication [29]. The E-value was calculated to consider the confounding strength capable of moving the observed association to any other value [30]. All analyses followed the intention-to-treat principle. SAS version 9.4 (SAS Institute, Cary, NC, USA), EZR version 1.55 (Saitama Medical Center, Jichi Medical University, Saitama, Japan), and R statistical software version 4.2.1 (R Group for Statistical Computing) with library fitSmoothHazard were used for all statistical analyses.

## Ethics

All enrollee data were anonymized to protect participant confidentiality [22]. This study adhered to the principles of the Declaration of Helsinki and was approved by the Medical Ethics Committee of Shizuoka Graduate School of Public Health in Shizuoka, Japan (#SGUPH_2021_001_031), and this committee waived the requirement for informed consent.

## Results

### Study population

After application of the inclusion and exclusion criteria (Fig 2), the final group consisted of 6860 patients in the suvorexant group and 50,203 patients in the BZRA group. Before propensity score matching, the patients in the suvorexant group were older and comprised a higher proportion of men. More comorbidities and concomitant medications were also observed (Table 1). During the follow-up period (average: 927.82 days), 1278 patients in the BZRA group and 259 patients in the suvorexant group developed hip fractures.

### Increment of fracture risk with orexin receptor antagonist prescriptions

Before propensity score matching, the suvorexant group had a higher risk of fracture than the BZRA group (HR, 1.37; 95% CI, 1.20–1.57) without adjustment for confounders.

After propensity score matching, all covariates were balanced between the two groups (Table 2). During the observational period, 259 of 6855 patients in the suvorexant group and

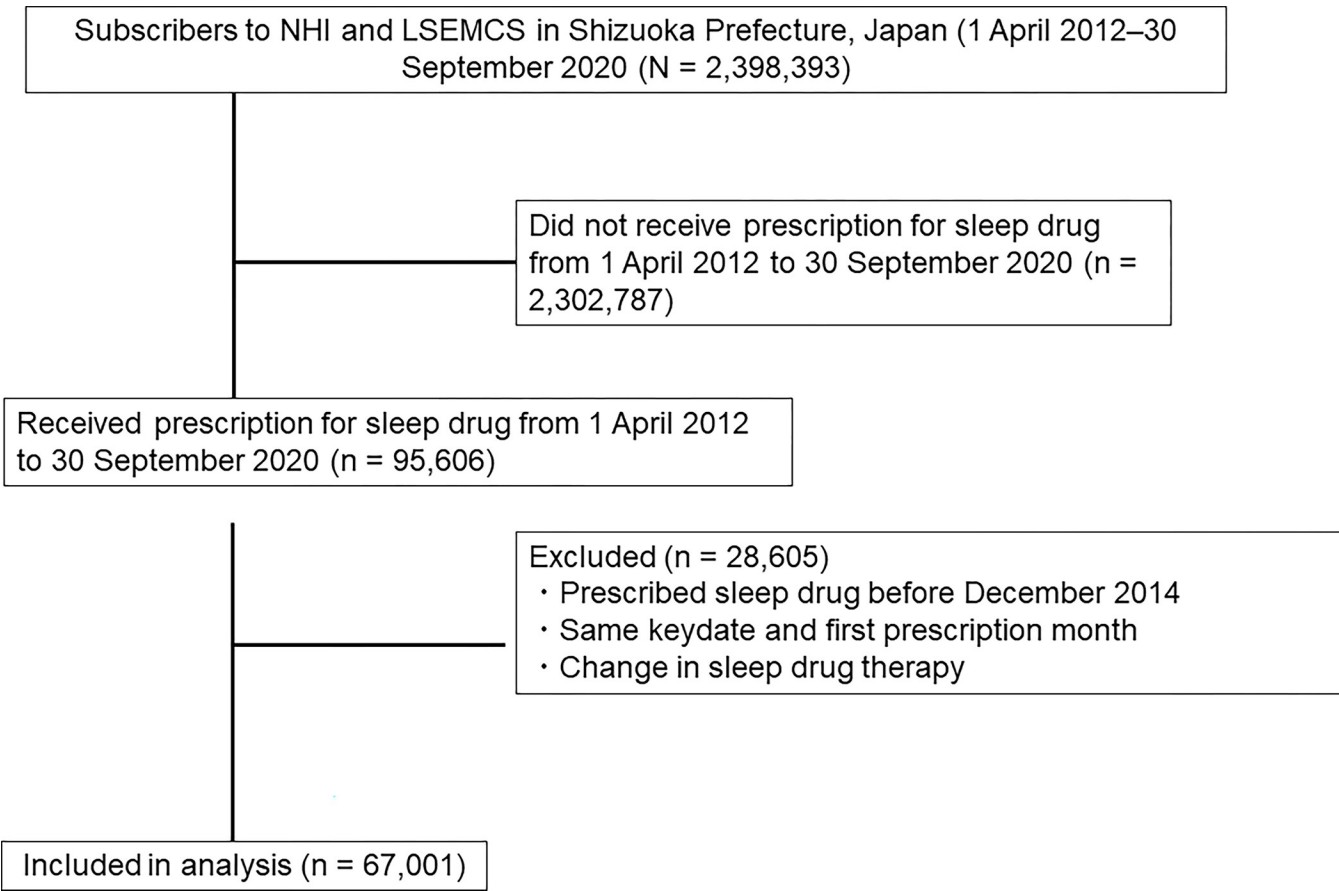

**Fig 2. Flow diagram of study cohort enrollment.** NHI: National Health Insurance, LSMSCE: Late-Stage Medical Care System for the Elderly.

249 of 6855 patients in the BZRA group developed hip fractures. The suvorexant group had a higher risk of fracture than the BZRA group (HR, 1.48; 95% CI, 1.20–1.82). The corresponding E-value (95% CI) was 2.31 (0.63–3.99). A further analysis of time-specific hazards showed that more fracture events occurred in the suvorexant group immediately after initiation of oral medication (Fig 3).

## Subgroup analysis

In a prespecified subgroup analysis (Fig 4), no significant differences were found according to age of >85 years, diabetes, neurological disease, renal failure, liver disease, treatment with alpha 1 blockers, and treatment with oral steroids; however, the results were similar in the other groups.

## Discussion

The present study showed that suvorexant increased the risk of fractures compared with BZRAs in a matched cohort using a large-scale database. To our knowledge, this study is the first to compare suvorexant with BZRA sleep drugs. A similar study that compared Z-drugs with suvorexant showed no difference in fracture risk between suvorexant and Z-drugs [31]. Unlike our study, only Z-drugs were compared; therefore, a simple comparison cannot be made [31]. In this study, suvorexant was prescribed more frequently than BZRAs to patients of

**Table 1. Baseline characteristics of BZRA group and suvorexant group before propensity score matching.**

| Variable | Category | Before matching | | P value |
|---|---|---|---|---|
| | | BZRA group | Suvorexant group | |
| | | n = 50,203 | n = 6860 | |
| Age | 65 to <75 years | 24,931 (49.7) | 2474 (36.1) | <0.001 |
| | 75 to <85 years | 18,174 (36.2) | 2639 (38.5) | |
| | 85 to <95 years | 6631 (13.2) | 1591 (23.2) | |
| | ≥95 years | 467 (0.9) | 156 (2.3) | |
| Sex | Male | 20,892 (41.6) | 3398 (49.5) | <0.001 |
| **Comorbidity** | | | | |
| Cardiac arrhythmias | Presence | 8867 (17.7) | 1415 (20.6) | <0.001 |
| Valvular disease | Presence | 3298 (6.6) | 625 (9.1) | <0.001 |
| Pulmonary circulation disorders | Presence | 175 (0.4) | 43 (0.6) | 0.001 |
| Hypertension | Presence | 31,343 (62.4) | 4575 (66.7) | <0.001 |
| Neurological disorders | Presence | 2443 (4.9) | 502 (7.3) | <0.001 |
| Diabetes | Presence | 4137 (8.2) | 739 (10.8) | <0.001 |
| Tumor | Presence | 8372 (16.7) | 1600 (23.3) | <0.001 |
| Hypothyroidism | Presence | 1432 (2.9) | 240 (3.5) | 0.004 |
| Renal failure | Presence | 2392 (4.8) | 464 (6.8) | <0.001 |
| Liver disease | Presence | 8663 (17.3) | 1193 (17.4) | 0.782 |
| Peptic ulcer disease excluding bleeding | Presence | 9573 (19.1) | 1290 (18.8) | 0.601 |
| Rheumatoid arthritis/collagen | Presence | 2264 (4.5) | 311 (4.5) | 0.929 |
| Coagulopathy | Presence | 939 (1.9) | 214 (3.1) | <0.001 |
| Obesity | Presence | 170 (0.3) | 25 (0.4) | 0.734 |
| Weight loss | Presence | 367 (0.7) | 71 (1.0) | 0.010 |
| Fluid and electrolyte disorders | Presence | 7303 (14.6) | 1223 (17.8) | <0.001 |
| Deficiency anemia | Presence | 4782 (9.5) | 840 (12.2) | <0.001 |
| Alcohol abuse | Presence | 299 (0.6) | 60 (0.9) | <0.009 |
| Drug abuse | Presence | 22 (0.04) | 4 (0.06) | 0.611 |
| Depression | Presence | 2541 (5.1) | 397 (5.8) | 0.012 |
| Psychoses | Presence | 925 (1.8) | 278 (4.1) | <0.001 |
| AIDS/HIV | Presence | 9 (0.02) | 1 (0.01) | 0.840 |
| Congestive heart failure | Presence | 8235 (16.4) | 1618 (23.6) | <0.001 |
| Peripheral vascular disorders | Presence | 6846 (13.6) | 995 (14.5) | 0.052 |
| Chronic pulmonary disorders | Presence | 12,367 (24.6) | 1792 (26.1) | 0.008 |
| Blood loss anemia | Presence | 180 (0.4) | 60 (0.9) | <0.001 |
| **Medication** | | | | |
| α1 receptor blocking drug | Presence | 16,315 (32.5) | 2238 (32.6) | 0.835 |
| Antihypertensive drug | Presence | 29,612 (59.0) | 4443 (64.8) | <0.001 |
| Diuretic drug | Presence | 5240 (10.4) | 1117 (16.3) | <0.001 |
| Antipsychotic drug | Presence | 1759 (3.5) | 369 (5.4) | <0.001 |
| Tricyclic antidepressant drug | Presence | 412 (0.8) | 207 (3.0) | <0.001 |
| Antihistamine drug | Presence | 17,607 (35.1) | 2195 (32.0) | <0.001 |
| Steroid | Presence | 11,285 (22.5) | 1450 (21.1) | 0.012 |
| Anti-receptor activator of NF-kB ligand drug | Presence | 944 (1.9) | 164 (2.4) | 0.005 |
| Bisphosphonate | Presence | 4451 (8.9) | 616 (9.0) | 0.757 |
| Calcium drug | Presence | 708 (1.4) | 94 (1.4) | 0.791 |
| Calcitonin | Presence | 785 (1.6) | 87 (1.3) | 0.055 |
| Ipriflavone | Presence | 7 (0.01) | 1 (0.01) | 0.967 |

*(Continued)*

**Table 1.** (Continued)

| Variable | Category | Before matching | | P value |
|---|---|---|---|---|
| | | BZRA group | Suvorexant group | |
| | | n = 50,203 | n = 6860 | |
| Parathyroid hormone | Presence | 475 (1.0) | 74 (1.1) | 0.299 |
| Selective estrogen receptor modulator | Presence | 1554 (3.1) | 178 (2.6) | 0.021 |
| Vitamin D | Presence | 5826 (11.6) | 801 (11.7) | 0.862 |

Data are presented as n (%). BZRA: benzodiazepine receptor agonist, AIDS: acquired immunodeficiency syndrome, HIV: human immunodeficiency virus, NF-kB: nuclear factor kappa B.

advanced age and patients with physical comorbidities (Table 1). Several variables were used to eliminate as much bias as possible, and a propensity score-matching approach was utilized. A recent survey of Japanese physicians showed that physicians who frequently prescribed orexin receptor antagonists were more concerned about safety when selecting a sleep medication than physicians who did not frequently prescribe such drugs [32]. Differences in attitudes toward prescribing were not adjusted for in this study and therefore remained a source of bias.

Several potential risk factors for fractures have been reported, and sleep drugs are one of them. The suvorexant group was expected to have a low risk of fracture because suvorexant is a newly launched sleep aid and has been shown to have a low risk of delirium. However, our results suggest that suvorexant may increase the risk of fracture. Furthermore, the incidence tended to be higher especially in the early phase of treatment (Fig 3), and caution may be required.

We considered two mechanisms underlying the increased risk of hip fracture risk in the suvorexant group. First, suvorexant has a weak sleep induction effect [33–35]. Therefore, the risk of falling due to somnolence is high. This may have contributed to the higher risk of hip fractures in the suvorexant group than in the BZRA group. Second, we know that the duration of REM sleep is longer with suvorexant than BZRAs, which leads to more nightmares [36]. We believe that an increased incidence of nightmares may increase the risk of falling at night.

In the subgroup analysis (Fig 4), there was no difference in the incidence of hip fracture between patients aged ≥85 years and those aged ≥95 years, with neuropathy, without hypertension, with diabetes, with kidney disease, taking alpha 1 blockers, and taking oral steroids. The exact reasons for these differences are unknown. However, we believe that these subgroups are almost exclusively high-risk groups for worse outcomes and that the reason for the differences may be the reduced risk of femoral fracture due to falls in both groups. For example, the fact that there is no difference in fracture risk among individuals aged >85 years is expected to indicate that healthcare providers and caregivers may be concerned about falls in all patients who are prescribed sleep-inducing drugs. It is also possible that the risk of falling is lower in patients aged ≥85 years because many of these patients are unable to stand due to decreased physical activity. Although the present study does not allow for further investigation, future studies that examine each of these subgroups may yield additional findings and insights.

This study had several limitations. First, residual unmeasured confounders may remain despite efforts to align the groups and eliminate selection bias using propensity score matching. However, we consider it unlikely that these unmeasured confounders would change the results when calculating E-values. Second, we could not collect information on family history, medication status, socioeconomic status, fracture mechanisms, or genetic data. Third, the diagnosis of hip fracture in this study was dependent on claims data, and we do not know whether the fracture data were over- or under-extracted from the database. Fourth, it is not yet

**Table 2. Baseline characteristics of BZRA group and suvorexant group after propensity score matching.**

| Variable (reference) | Category | After matching | | SMD |
|---|---|---|---|---|
| | | BZRA group | Suvorexant group | |
| | | n = 6855 | n = 6855 | |
| Age | 65 to <75 years | 2376 (34.7) | 2472 (36.1) | 0.007 |
| | 75 to <85 years | 2811 (41.0) | 2635 (38.5) | |
| | 85 to <95 years | 1504 (22.0) | 1586 (23.2) | |
| | ≥95 years | 167 (2.4) | 156 (2.3) | |
| Sex (female) | Male | 3334 (48.7) | 3391 (49.5) | 0.008 |
| **Comorbidity** | | | | |
| Cardiac arrhythmias | Presence | 1384 (20.2) | 1411 (20.6) | 0.004 |
| Valvular disease | Presence | 595 (8.7) | 622 (9.1) | 0.004 |
| Pulmonary circulation disorders | Presence | 48 (0.7) | 42 (0.6) | 0.001 |
| Hypertension | Presence | 4610 (67.3) | 4566 (66.7) | 0.006 |
| Neurological disorders | Presence | 510 (7.4) | 499 (7.3) | 0.002 |
| Diabetes | Presence | 739 (10.8) | 739 (10.8) | <0.001 |
| Tumor | Presence | 1533 (22.4) | 1592 (23.2) | 0.009 |
| Hypothyroidism | Presence | 195 (2.8) | 234 (3.4) | 0.006 |
| Renal failure | Presence | 432 (6.3) | 462 (6.7) | 0.004 |
| Liver disease | Presence | 1140 (16.6) | 1189 (17.4) | 0.007 |
| Peptic ulcer disease excluding bleeding | Presence | 1242 (18.1) | 1284 (18.7) | 0.006 |
| Rheumatoid arthritis/collagen | Presence | 278 (4.1) | 310 (4.5) | 0.005 |
| Coagulopathy | Presence | 205 (3.0) | 213 (3.1) | 0.001 |
| Obesity | Presence | 28 (0.4) | 25 (0.4) | <0.001 |
| Weight loss | Presence | 63 (0.9) | 71 (1.0) | 0.001 |
| Fluid and electrolyte disorders | Presence | 1165 (17.0) | 1221 (17.8) | 0.008 |
| Deficiency anemia | Presence | 817 (11.9) | 835 (12.2) | 0.003 |
| Alcohol abuse | Presence | 66 (1.0) | 60 (0.9) | 0.003 |
| Depression | Presence | 397 (5.8) | 394 (5.8) | <0.001 |
| Psychoses | Presence | 264 (3.9) | 274 (4.0) | 0.002 |
| Congestive heart failure | Presence | 1560 (22.8) | 1613 (23.6) | 0.008 |
| Peripheral vascular disorders | Presence | 954 (13.9) | 994 (14.5) | 0.006 |
| Chronic pulmonary disorders | Presence | 1748 (25.5) | 1788 (26.1) | 0.006 |
| Blood loss anemia | Presence | 63 (0.9) | 58 (0.8) | 0.001 |
| **Medication** | | | | |
| α1 receptor blocking drug | Presence | 2175 (31.8) | 2234 (32.6) | 0.009 |
| Antihypertensive drug | Presence | 4446 (64.9) | 4432 (64.7) | 0.002 |
| Diuretic drug | Presence | 1092 (15.9) | 1110 (16.2) | 0.003 |
| Antipsychotic drug | Presence | 357 (5.2) | 364 (5.3) | 0.001 |
| Tricyclic antidepressant drug | Presence | 203 (3.0) | 198 (2.9) | 0.005 |
| Antihistamine drug | Presence | 2173 (31.7) | 2190 (32.0) | 0.003 |
| Steroid | Presence | 11407 (166.5) | 1446 (21.1) | 0.006 |
| Anti-receptor activator of NF-kB ligand drug | Presence | 178 (2.6) | 164 (2.4) | 0.002 |
| Bisphosphonate | Presence | 609 (8.9) | 612 (8.9) | <0.001 |
| Calcium drug | Presence | 85 (1.2) | 94 (1.4) | 0.001 |
| Calcitonin | Presence | 87 (1.3) | 87 (1.3) | <0.001 |
| Parathyroid hormone | Presence | 66 (1.0) | 74 (1.1) | 0.002 |
| Selective estrogen receptor modulator | Presence | 175 (2.6) | 177 (2.6) | <0.001 |

(*Continued*)

**Table 2.** (Continued)

| Variable (reference) | Category | After matching | | SMD |
| --- | --- | --- | --- | --- |
| | | **BZRA group** | **Suvorexant group** | |
| | | **n = 6855** | **n = 6855** | |
| Vitamin D | Presence | 778 (11.4) | 800 (11.7) | 0.003 |

Data are presented as n (%). BZRA: benzodiazepine receptor agonist, SMD: standardized mean difference, NF-kB: nuclear factor kappa B.

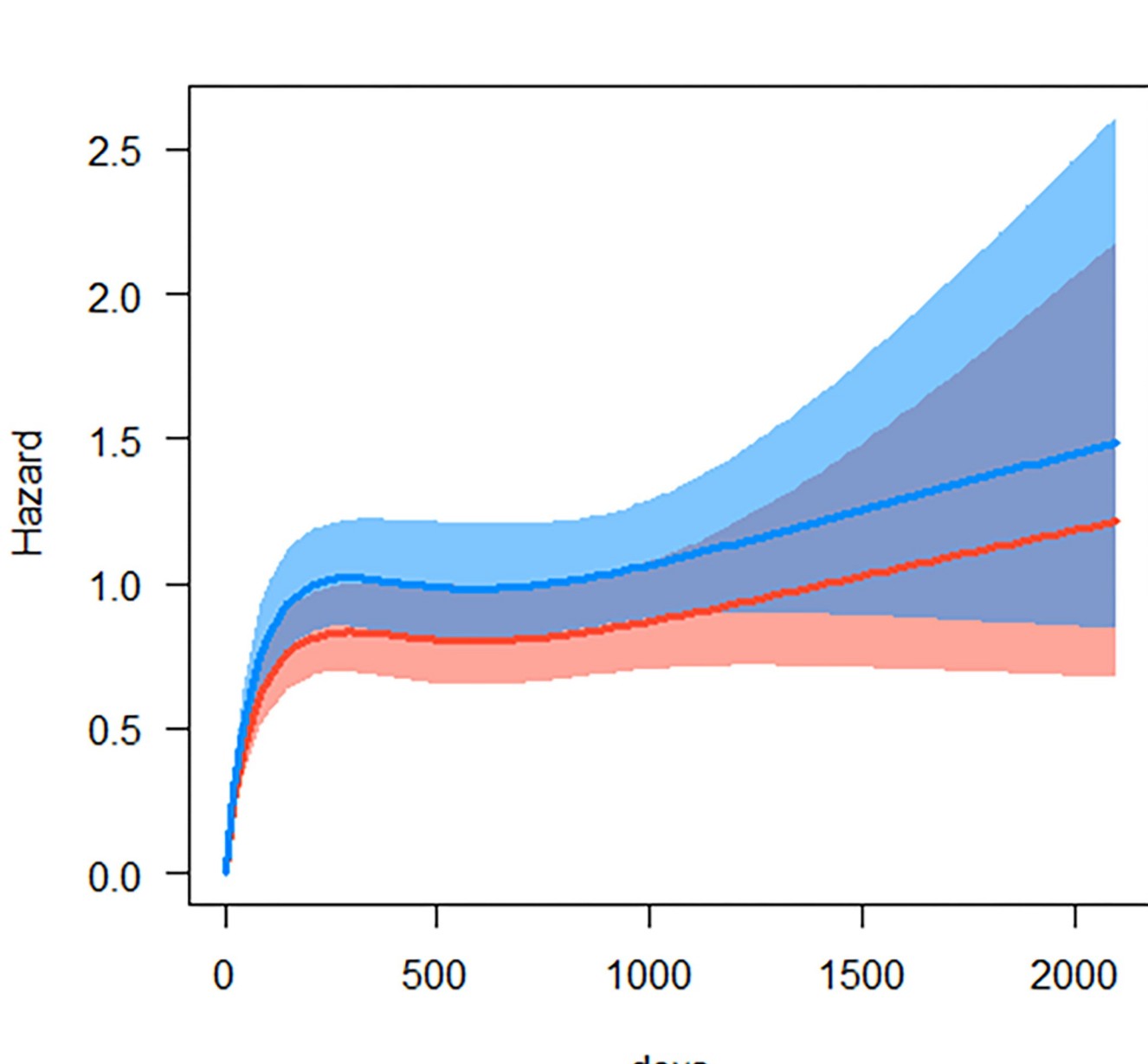

**Fig 3. Time-specific hazard estimates.** The BZRA and suvorexant groups had a higher risk of hip fracture events immediately after initiation of oral therapy, with the suvorexant group having a higher risk than the BZRA group. BZRA: benzodiazepine receptor agonist.

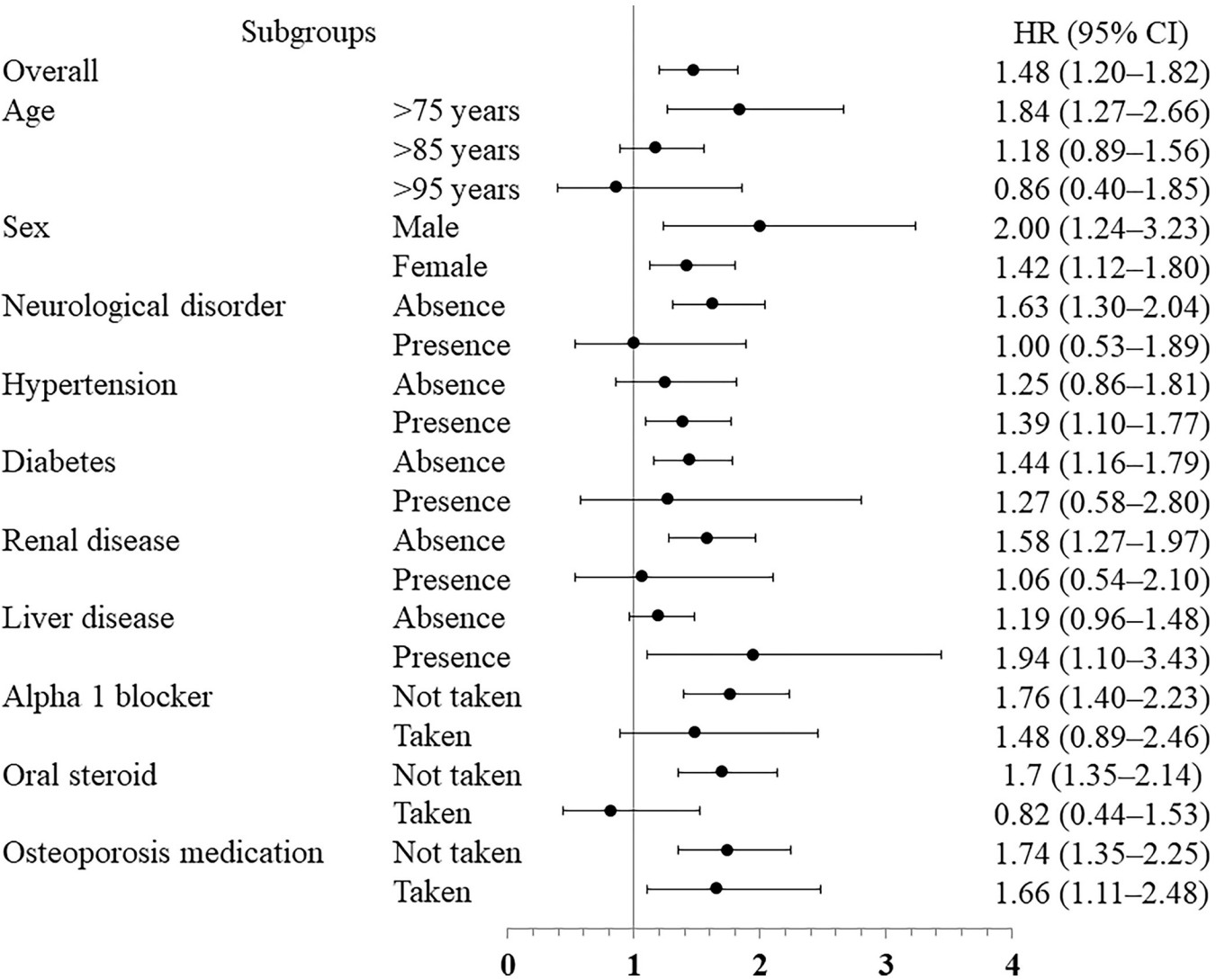

**Fig 4. Results of subgroup analysis.** HR: hazard ratio, CI: confidence interval, BZRA: benzodiazepine receptor agonist.

possible to assess capacity–response relationships because of the inability to obtain accurate information on the dose and duration of sleep-inducing drug prescriptions. Fifth, this study did not analyze lemborexant, the most recently developed orexin receptor antagonist, because of the observation period. Lemborexant has a higher sleep induction effect than suvorexant [37], and its inclusion in the analysis may have eliminated the disadvantages of orexin receptor antagonists. Despite the above limitations, our study findings suggest that suvorexant, a novel sleep drug, can be associated with an increased risk of fractures. Future randomized controlled trials and large prospective studies are needed to confirm the results of this study.

## Conclusion

Analysis of a population-based cohort treated with sleep drugs revealed that the risk of hip fracture can be higher when suvorexant, a novel sleep drug, is used than when BZRA sleep drugs are used.

## Supporting information

**S1 Table. Sleep drugs included in each drug category.**
(DOCX)

**S2 Table. Definitions of comorbidities using ICD-10 codes.** ICD-10: International Classification of Diseases, 10th Revision, AIDS: acquired immunodeficiency syndrome, HIV: human immunodeficiency virus.
(DOCX)

## Acknowledgments

A database from the Japan Pharmaceutical Information Center was used for the drug code search. The authors thank Angela Morben, DVM, ELS, from Edanz (https://jp.edanz.com/ac) for editing a draft of this manuscript.

## Author Contributions

**Conceptualization:** Ryozo Yoshioka, Seiichiro Yamamoto, Eiji Nakatani.

**Data curation:** Eiji Nakatani.

**Formal analysis:** Ryozo Yoshioka.

**Funding acquisition:** Seiichiro Yamamoto.

**Investigation:** Ryozo Yoshioka, Seiichiro Yamamoto, Eiji Nakatani.

**Methodology:** Ryozo Yoshioka, Seiichiro Yamamoto, Eiji Nakatani.

**Project administration:** Ryozo Yoshioka.

**Resources:** Eiji Nakatani.

**Software:** Seiichiro Yamamoto, Eiji Nakatani.

**Supervision:** Seiichiro Yamamoto, Eiji Nakatani.

**Validation:** Eiji Nakatani.

**Visualization:** Ryozo Yoshioka.

**Writing – original draft:** Ryozo Yoshioka.

**Writing – review & editing:** Seiichiro Yamamoto, Eiji Nakatani.

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
