## [Decision Letter · Decision Letter 0]

14 Feb 2023

PONE-D-23-01072Effectiveness of suvorexant versus benzodiazepine receptor agonist sleep drugs in reducing the risk of hip fracture: findings from a regional population-based cohort studyPLOS ONE

Dear Dr. Yoshioka,

Thank you for submitting your manuscript to PLOS ONE. After careful consideration, we feel that it has merit but does not fully meet PLOS ONE’s publication criteria as it currently stands. Therefore, we invite you to submit a revised version of the manuscript that addresses the points raised during the review process.

We look forward to receiving your revised manuscript.

Kind regards,

Norio Yasui-Furukori

Academic Editor

PLOS ONE

“The Shizuoka Graduate University of Public Health conducts contract research projects for public health in Shizuoka Prefecture, including the current study, and funding for this work was provided by Shizuoka Prefecture.”

Reviewers' comments:

Reviewer's Responses to Questions

**Comments to the Author**

1. Is the manuscript technically sound, and do the data support the conclusions?

Reviewer #1: Yes

Reviewer #2: Partly

2. Has the statistical analysis been performed appropriately and rigorously? 

Reviewer #1: Yes

Reviewer #2: I Don't Know

3. Have the authors made all data underlying the findings in their manuscript fully available?

Reviewer #1: Yes

Reviewer #2: Yes

4. Is the manuscript presented in an intelligible fashion and written in standard English?

Reviewer #1: Yes

Reviewer #2: No

5. Review Comments to the Author

Reviewer #1: strength :the present study showed that suvorexant increased the risk of fractures compared with BZRAs in a matched cohort using a large-scale database. This study is the first to compare suvorexant with BZRA sleep drugs.

This draft have merit for publication.

Reviewer #2: I would like to thank the authors for the tremendous effort they put into this study. The findings from this study are important because many physicians may be interested in whether suvorexant has a lower risk of hip fracture compared to benzodiazepine receptor agonists. However, this study may have methodological issues that should be clarified. In addition, there is a need for more in-depth discussion of the findings from this study.

Because this is a database-based study, it is not clear whether patients prescribed Suvorexant were actually taking it internally. Therefore, I propose to change the term "newly prescribed Suvorexant" to "newly prescribed Suvorexant" rather than "newly taking Suvorexant".

(Lines 37– 39)

Because this study is not an RCT and confounding factors cannot be completely eliminated even if various factors are adjusted, it is not possible to conclude that suvorexant increases fracture risk compared to benzodiazepine receptor agonists. A more modest statement about the conclusion should be made.

(Lines 56– 57)

It is not all sleep drugs that increase the risk of delirium and fractures, but benzodiazepine receptor agonists, correct? Please revise your statement appropriately and add a citation.

(Lines 72– 79)

Based on the fact that suvorexant is not muscle relaxant and does not induce delirium, the authors may have hypothesized that suvorexant has a lower fracture risk than benzodiazepine receptor agonists. Please clearly indicate why the authors conducted this study.

(Introduction)

In the introduction section, the authors mentioned the risk of delirium with benzodiazepine receptor agonists. However, the authors did not mention the relationship between delirium and hip fractures, which may make it difficult for the reader to understand what the authors are trying to convey in the introduction. Therefore, please add whether delirium increases the risk of hip fractures. Also mention that it has been suggested that suvorexant may be effective in preventing delirium and that it is generally believed that suvorexant does not increase the risk of delirium.

(Lines 83– 96)

Describe in detail the information held by SKDB. Does it have information such as the name of the drug prescribed, the date it was prescribed, the amount and dosage of the drug prescribed, and the number of days the drug was prescribed?

(Lines 105– 110)

Please describe in detail the eligibility criteria and exclusion criteria. Were patients who were prescribed benzodiazepine receptor agonists and suvorexant as sleep medications for the first time at the same time excluded? Also, how did you treat patients whose first prescribed sleep drug was suvorexant, but who were switched or added to a benzodiazepine receptor agonist along the way? Were such patients censored at the time they were switched to or added to benzodiazepine receptor agonists? Further, how were patients treated who discontinue their first prescribed sleeping medication before the end of observation?

(Lines 117– 118)

Please list the name of the drug in Table S1. Codes alone will not be understood by non-Japanese readers.

(Discussion)

As shown in Table 1, suvorexant was more commonly prescribed to the elderly and patients with physical comorbidities than benzodiazepine receptor agonists. Discuss this as it appears to be an important finding.

A recent survey of Japanese physicians indicates that physicians who frequently prescribed orexin receptor antagonists are more concerned about safety than those who did not when choosing a sleep medication.

https://www.frontiersin.org/articles/10.3389/fpsyt.2023.1071962/abstract

(Lines 228– 229)

Cite previous studies that reported that a weak effect on difficulty falling asleep increased the risk of falls. Also, I assume that most of the previous studies are on benzodiazepine receptor agonists. Are there any studies comparing the effect on difficulty falling asleep and the risk of falls among orexin receptor antagonists?

(Lines 234– 241)

The discussion on subgroup analysis is too simplistic. Please discuss more about the results of the subgroup analysis, at least with respect to age.

6. PLOS authors have the option to publish the peer review history of their article (what does this mean?). If published, this will include your full peer review and any attached files.

Reviewer #1: **Yes: **Jun-Jun Yeh

Reviewer #2: **Yes: **Masahiro Takeshima

---

## [Author Response · Author response to Decision Letter 0]

31 Mar 2023

Response to the comments of Reviewer #1

1. strength :the present study showed that suvorexant increased the risk of fractures compared with BZRAs in a matched cohort using a large-scale database. This study is the first to compare suvorexant with BZRA sleep drugs. This draft have merit for publication.

Response: We appreciate this favorable response from the reviewer.

Response to the comments of Reviewer #2

We wish to express our appreciation to the reviewer for the insightful comments, which have helped us significantly improve our paper.

1. Because this is a database-based study, it is not clear whether patients prescribed Suvorexant were actually taking it internally. Therefore, I propose to change the term "newly prescribed Suvorexant" to "newly prescribed Suvorexant" rather than "newly taking Suvorexant".

Response: We thank the reviewer for this comment. Because this was a database-based study, we have changed “newly taking suvorexant” to “newly prescribed suvorexant” as the reviewer pointed out. This change has been made throughout the manuscript.

2. (Lines 37– 39) Because this study is not an RCT and confounding factors cannot be completely eliminated even if various factors are adjusted, it is not possible to conclude that suvorexant increases fracture risk compared to benzodiazepine receptor agonists. A more modest statement about the conclusion should be made.

Response: We thank the reviewer for this very helpful comment. We have accordingly revised the statement to provide a more modest conclusion.

“In the Japanese regional population who used sleep drugs, patients taking suvorexant were at higher risk of hip fracture than patients taking BZRAs.” (Lines 40-46)

->

“Among people in the Japanese regional population who used sleep drugs, patients taking suvorexant can be at higher risk of hip fracture than patients taking BZRAs.”

“…. revealed a higher risk of hip fracture when suvorexant, a novel sleep drug, was used than when BZRA sleep were used.” (Lines 290-292)

->

“…. revealed that the risk of hip fracture can be higher when suvorexant, a novel sleep drug, is used than when BZRA sleep drugs are used.”

3. (Lines 56– 57) It is not all sleep drugs that increase the risk of delirium and fractures, but benzodiazepine receptor agonists, correct? Please revise your statement appropriately and add a citation. 

Response: We thank the reviewer for this very accurate remark. We have accordingly revised the text as follows.

“However, sleep drugs have side effects such as delirium and hip fractures.” (Line 60)

->

“However, BZRAs have side effects such as delirium and hip fractures.”

4. (Lines 72– 79) Based on the fact that suvorexant is not muscle relaxant and does not induce delirium, the authors may have hypothesized that suvorexant has a lower fracture risk than benzodiazepine receptor agonists. Please clearly indicate why the authors conducted this study. 

Response: We thank the reviewer for this very valid point. We hypothesized that differences in muscle relaxation and delirium may suggest a risk of fracture with suvorexant. Therefore, we have revised the text as shown in our response to Comment 5 below.

5. (Introduction) In the introduction section, the authors mentioned the risk of delirium with benzodiazepine receptor agonists. However, the authors did not mention the relationship between delirium and hip fractures, which may make it difficult for the reader to understand what the authors are trying to convey in the introduction. Therefore, please add whether delirium increases the risk of hip fractures. Also mention that it has been suggested that suvorexant may be effective in preventing delirium and that it is generally believed that suvorexant does not increase the risk of delirium. 

Response: We thank the reviewer for providing this accurate information. We have added references on the reduction in the risk of delirium with suvorexant.

“…and have a low risk of delirium because of their action of inhibiting wakefulness [18-20]. However, because they are novel drugs, their clinical significance remains to be evaluated, particularly concerning falls and hip fractures.

 In the present study, we investigated whether the orexin receptor antagonist suvorexant, a novel sleep drug, reduces the risk of hip fractures compared with BZRA sleep drugs.” (Lines 78–86)

->

“… and have a low risk of delirium because of their action of inhibiting wakefulness [18-20]. Recent studies have also indicated a preventive effect of suvorexant against delirium [19,20]. Prior studies have shown an increased risk of fracture with delirium [21], and we hypothesize that these findings suggest that suvorexant is associated with a higher risk of fracture compared with BZRAs. However, because they are novel drugs, their clinical significance remains to be evaluated, particularly concerning falls and hip fractures. Because suvorexant is not a muscle relaxant and does not induce delirium, we investigated whether the orexin receptor antagonist suvorexant, a novel sleep drug, reduces the risk of hip fractures compared with BZRA sleep drugs.”

6. (Lines 83– 96) Describe in detail the information held by SKDB. Does it have information such as the name of the drug prescribed, the date it was prescribed, the amount and dosage of the drug prescribed, and the number of days the drug was prescribed? 

Response: We thank the reviewer for this helpful question. We have added the following explanation of the SKDB to the Methods section.

“The SKDB maintains data on the name of the prescribed drug, prescription date, prescription drug dosage, and number of prescription days.” (Lines 104–106)

7. (Lines 105– 110) Please describe in detail the eligibility criteria and exclusion criteria. Were patients prescribed benzodiazepine receptor agonists and suvorexant as sleep medications for the first time at the same time excluded? Also, how did you treat patients whose first prescribed sleep drug was suvorexant, but who were switched or added to a benzodiazepine receptor agonist along the way? Were such patients censored at the time they were switched to or added to benzodiazepine receptor agonists? Further, how were patients treated who discontinue their first prescribed sleeping medication before the end of observation?

Response: We thank the reviewer for these critical questions. We conducted this study with an intent-to-treat analysis. Therefore, drug changes or interruptions during the follow-up period were not considered. We have added the analysis method and inclusion and exclusion criteria to the Methods section.

“The inclusion criterion was new prescription of sleep drugs during the above-described study period. The exclusion criteria were new prescription of sleep drugs before 2014, fracture outcome and sleep drug prescription occurring in the same month, and age of <65 years. We conducted this study with an intent-to-treat analysis.” (Lines 121–124)

8. (Lines 117– 118) Please list the name of the drug in Table S1. Codes alone will not be understood by non-Japanese readers. 

Response: We thank the reviewer for raising this essential point. We have accordingly revised the table as shown below.

S1 Table. Definitions of sleep drugs using Japanese drug codes

Sleep drug Japanese drug code

Benzodiazepines 1124001 1124002 1124003 1124005 1124007 1124008 1124009 1124010 1124013 1124014 1124015 1124017 1124019 1124020 1124021 1124022 1124023 1124024 1124025 1124027 1124028 1124029 1124030 1129006

Benzodiazepine receptor agonists 1129007 1129009 1129010 

Orexin receptor agonist 1190023

->

S1 Table. Sleep drugs included in each drug category

Sleep drug categories Sleep drugs

Benzodiazepines Estazolam, flurazepam, nitrazepam, haloxazolam, triazolam, flunitrazepam, brotizolam, lormetazepam, oxazolam, cloxazolam, clorazepate, diazepam, fludiazepam, bromazepam, medazepam, lorazepam, alprazolam, flutazolam, mexazolam, flutrazepam, chlordiazepoxide, loflazepate, quazepam, rilmazafone

Benzodiazepine receptor agonists Zolpidem, zopiclone, eszopiclone 

Orexin receptor agonist Suvorexant

9. (Discussion) As shown in Table 1, suvorexant was more commonly prescribed to the elderly and patients with physical comorbidities than benzodiazepine receptor agonists. Discuss this as it appears to be an important finding. A recent survey of Japanese physicians indicates that physicians who frequently prescribed orexin receptor antagonists are more concerned about safety than those who did not when choosing a sleep medication. https://www.frontiersin.org/articles/10.3389/fpsyt.2023.1071962/abstract

Response: We thank the reviewer for raising these essential points and providing a reference. We have added the following information to the revised manuscript (Lines 238–245).

“In this study, suvorexant was prescribed more frequently than BZRAs to patients of advanced age and patients with physical comorbidities (Table 1). Several variables were used to eliminate as much bias as possible, and a propensity score-matching approach was utilized. A recent survey of Japanese physicians showed that physicians who frequently prescribed orexin receptor antagonists were more concerned about safety when selecting a sleep medication than physicians who did not frequently prescribe such drugs [32]. Differences in attitudes toward prescribing were not adjusted for in this study and therefore remained a source of bias.”

10. (Lines 228– 229) Cite previous studies that reported that a weak effect on difficulty falling asleep increased the risk of falls. Also, I assume that most of the previous studies are on benzodiazepine receptor agonists. Are there any studies comparing the effect on difficulty falling asleep and the risk of falls among orexin receptor antagonists? 

Response: We thank the reviewer for these critical remarks. It is a clinical hypothesis that the inability to fall asleep while taking sleep drugs is associated with a risk of falls and fractures. However, we found no references regarding the relationship between a risk of bone fractures and difficulty falling asleep.

11. (Lines 234– 241) The discussion on subgroup analysis is too simplistic. Please discuss more about the results of the subgroup analysis, at least with respect to age.

Response: We thank the reviewer for this helpful suggestion. We have revised the following text (Lines 264–272).

“For example, we believe that in these subgroups, health care providers may have been more attentive to episodes such as falls or patients may have been less active”

->

“For example, the fact that there is no difference in fracture risk among individuals aged >85 years is expected to indicate that healthcare providers and caregivers may be concerned about falls in all patients who are prescribed sleep-inducing drugs. It is also possible that the risk of falling is lower in patients aged ≥85 years because many of these patients are unable to stand due to decreased physical activity. Although the present study does not allow for further investigation, future studies that examine each of these subgroups may yield additional findings and insights.”

We believe that the above responses fully address the reviewers’ concerns.

We again thank the reviewers for their comments regarding our paper. We hope that our revised manuscript is now suitable for publication.

---

## [Editor Report · Decision Letter 1]

6 Apr 2023

Effectiveness of suvorexant versus benzodiazepine receptor agonist sleep drugs in reducing the risk of hip fracture: findings from a regional population-based cohort study

PONE-D-23-01072R1

Dear Dr. Nakatani,

We’re pleased to inform you that your manuscript has been judged scientifically suitable for publication and will be formally accepted for publication once it meets all outstanding technical requirements.

Kind regards,

Norio Yasui-Furukori

Academic Editor

PLOS ONE
---

## [Editor Report · Acceptance letter]

14 Apr 2023

PONE-D-23-01072R1 

Effectiveness of suvorexant versus benzodiazepine receptor agonist sleep drugs in reducing the risk of hip fracture: findings from a regional population-based cohort study 

Dear Dr. Nakatani:

I'm pleased to inform you that your manuscript has been deemed suitable for publication in PLOS ONE. Congratulations! Your manuscript is now with our production department. 

Kind regards, 

on behalf of

Dr. Norio Yasui-Furukori 

Academic Editor

PLOS ONE